# Methanol Mitigation during Manufacturing of Fruit Spirits with Special Consideration of Novel Coffee Cherry Spirits

**DOI:** 10.3390/molecules26092585

**Published:** 2021-04-28

**Authors:** Patrik Blumenthal, Marc C. Steger, Daniel Einfalt, Jörg Rieke-Zapp, Andrès Quintanilla Bellucci, Katharina Sommerfeld, Steffen Schwarz, Dirk W. Lachenmeier

**Affiliations:** 1Coffee Consulate, Hans-Thoma-Strasse 20, 68163 Mannheim, Germany; patrik.blumenthal@live.de (P.B.); marcsteger2@googlemail.com (M.C.S.); joerg.rieke_zapp@yahoo.de (J.R.-Z.); schwarz@coffee-consulate.com (S.S.); 2Yeast Genetics and Fermentation Technology, Institute of Food Science and Biotechnology, University of Hohenheim, Garbenstrasse 23, 70599 Stuttgart, Germany; daniel.einfalt@uni-hohenheim.de; 3Finca La Buena Esperanza, Pasaje Senda Florida Norte 124, San Salvador, El Salvador; coffeelbe@gmail.com; 4Chemisches und Veterinäruntersuchungsamt (CVUA) Karlsruhe, Weissenburger Strasse 3, 76187 Karlsruhe, Germany; katharina.sommerfeld@cvuaka.bwl.de

**Keywords:** alcoholic beverages, spirits, methanol, risk mitigation, legal limits, quality control

## Abstract

Methanol is a natural ingredient with major occurrence in fruit spirits, such as apple, pear, plum or cherry spirits, but also in spirits made from coffee pulp. The compound is formed during fermentation and the following mash storage by enzymatic hydrolysis of naturally present pectins. Methanol is toxic above certain threshold levels and legal limits have been set in most jurisdictions. Therefore, the methanol content needs to be mitigated and its level must be controlled. This article will review the several factors that influence the methanol content including the pH value of the mash, the addition of various yeast and enzyme preparations, fermentation temperature, mash storage, and most importantly the raw material quality and hygiene. From all these mitigation possibilities, lowering the pH value and the use of cultured yeasts when mashing fruit substances is already common as best practice today. Also a controlled yeast fermentation at acidic pH facilitates not only reduced methanol formation, but ultimately also leads to quality benefits of the distillate. Special care has to be observed in the case of spirits made from coffee by-products which are prone to spoilage with very high methanol contents reported in past studies.

## 1. Introduction

Methanol is an alcohol that is typically found in almost all kinds of alcoholic beverages and some other fermented food products [1,2,3,4,5]. Methanol may occur in alcoholic beverages through two major pathways: a natural one (pectin degradation), as well as an artificial one (adulteration by illegal addition of the pure compound). Only the latter pathway (adulteration) is typically associated with major morbidity and mortality due to methanol poisoning [6,7,8,9]. While adulteration is still prevalent and incidences have increased due to alcohol shortages during the COVID-19 pandemic [10], this article will exclusively focus on the first pathway, the natural content of methanol in spirits and its mitigation. Regarding the mitigation of problems related to methanol addition, we have recently provided a separate review [11].

In the human body, methanol may be endogenously present in low concentrations [12,13], while in most alcoholic beverages such as beer and wine, the natural content of methanol is also quite low. This differs with fruit spirits, so that the major focus on methanol reduction measures lies on this kind of beverage.

Spirits are alcoholic beverages that use fruits or other sugar-containing plant parts as the raw material. They are produced by alcoholic fermentation followed by distillation [14]. In Central European countries and in Russia, but also in Asia and many American countries, home production or artisanal small-scale production of spirits has a long tradition, while typically the sugar-containing materials of the region are preferred. For instance, countries in Central Europe mainly utilize fruits such as cherries, apples, and plums while other regions focus on grains (Eastern Europe) or sugar cane materials (Central and Southern America). From all natural materials used for fermentation, fruits are associated with the highest concentrations of methanol in the end-product, because of their pectin content. Typically, stone fruits of the genus *Prunus* (cherries, plums) and pome fruits of the genera *Malus* and *Pyrus* (apples, pears) are associated with the highest methanol levels. More recently, coffee cherries (genus *Coffea*) were identified as fruits possibly leading to comparably high methanol levels in their spirits [15].

Methanol concentrations in spirits are closely linked to enzymatic activities in the fruits and during the alcoholic fermentation process. Pectin methylesterase activity (1) may derive endogenously from the fruits themselves but also during alcoholic fermentation by pectin methylesterase formed from yeast metabolism or from other microorganisms [16,17,18]. Pectin methylesterase activity may also be exogenously introduced by addition of certain pectolytic enzyme preparations. A negligible pathway may be thermic demethylation of pectins [19].
(1)$${\mathrm{Pectin}+H}_{2}O\;\overset{\mathrm{Pectin}\;\mathrm{methylesterase}}{\to }\mathrm{Pectic}\;\mathrm{acid}+\mathrm{methanol}$$

When methanol has been released from the fruits’ pectin, it inevitably becomes part of the mash [20]. Its level is dependent on the degree of esterification of the pectin inside the fruits and the fruit-dependent ratio between sugar and pectin [5,21]. Another pathway suggested for methanol formation in protein-rich fruits such as jejube (Chinese date, *Ziziphus jujube* Mill.) was glycine deamination, followed by decarboxylation and reaction with nitrite from fertilizer use [22].

The European Union (EU) regulates maximum methanol contents in spirits dependent on the utilized raw materials [4,23,24]. For ethyl alcohol of agricultural origin, the maximum level of methanol is 30 g/hL of 100% vol alcohol (pure alcohol, pa), while for vodka it is 10 g/hL pa and the lowest level is defined for London gin with 5 g/hL pa. The limits are higher for fruit-based materials: for wine spirit 200 g/hL pa, for grape marc and cider 1000 g/hL pa, for fruit marc 1500 g/hL pa, for fruit spirits in general 1000 g/hL pa, except 1200 g/hL pa for apples, apricots, plum, mirabelle, peach, pear, blackberry and raspberry, and 1350 g/hL pa for quince, Williams pear and some other berries [23]. While these EU limits are set to reduce toxic effects on the human body, they were also judged as being rather low and, for some types of fruit, as challenging to be upheld by small artisanal distillers [25]. Lower limits in other countries such as the USA may also prohibit export of fruit spirits to these countries [26].

This article will review the possibilities to control and reduce the methanol content in fruit spirts and also describe some initial observations for the novel spirit made from coffee cherries.

## 2. Materials and Methods

A database research in January 2021 was conducted in Google Scholar and PubMed using the keyword combination “methanol, “reduction” and “spirits” or “alcoholic beverages”. It became quickly evident that the indexed international literature contains only few references about the topic. For that reason, the paper collection of the authors was screened for the key words to identify the gray literature mostly in German language industry magazines. The reference lists of all identified articles were screened for missing references. A narrative review was compiled from the available evidence.

## 3. Toxicity of Methanol in Alcoholic Beverages

Methanol is a colorless liquid and it is highly flammable. It is the simplest alcohol with a wide range of industrial applications. Methanol is also a natural ingredient in alcoholic beverages and spirits. To ensure that the residual methanol content present in spirits is safe, methanol content has to be strictly monitored [2,27].

Methanol is one of the few compounds occurring in foods for which excellent human toxicity data is available. This data mostly origins from the experience with poisonings from methanol containing spirits that sadly still regularly occur worldwide in connection with unrecorded and illicit alcohol consumption [11]. Methanol is metabolized in the body to its toxic metabolites, formaldehyde and formic acid. The accumulation of formic acid may cause metabolic acidosis including damage to the retina, the central nervous system and other organs [2,28].

It must be directly noted that such poisonings typically occur from methanol addition to spirits (mostly found on the illicit market), while the natural content due to fermentation from fruits does not typically exceed levels causing acute toxicity [11].

Poisoning outbreaks were reported from all regions worldwide, the size of which ranging from a few to over 800 victims, with fatality rates of over 30% in some instances [29].

Paine and Dayan [2] reported that the low concentrations of methanol naturally occurring in most alcoholic beverages are not causing any harm. According to WHO [29], methanol concentration in typical ranges of 6–27 mg/L in beer and 10–220 mg/L in spirits are not harmful. Paine and Dayan [2] also reported that the daily tolerable, virtually safe dose of methanol for an adult is 2 g and the toxic dose is 8 g. For a drinking volume of 100 mL of a spirit at 40% vol, the tolerable concentration would be 2% vol methanol (i.e., 5000 g/hL pa). Hence, the EU general limit for naturally occurring methanol in fruit spirits of 1000 g/hL pa [23] offers a safety margin of about 5 for heavy consumers of fruit spirits. Compared to other toxic food constituents, this margin is rather low, so that the limits must be strictly controlled and adhered by industry. Considering the demand for precautionary public health protection, it is obviously prudent to lower the methanol content in fruit spirits as low as it is reasonably achievable (ALARA principle).

## 4. Factors Influencing the Methanol Content of Fruit Spirits

Table 1 provides an overview of the major methods and approaches to reduce methanol in spirits. From their experience in practical work in spirits drinks control and distillation technology, the authors also provide a judgement about the applicability of the approaches, considering practical as well as economical aspects. The following sections are considering each approach in more detail.

### 4.1. Raw Materials, Mash Preparation and Fermentation

Prior to sensitization of industry regarding the methanol problem and the implementation of maximum limits by the EU in the first spirits regulation in 1989 [50], so-called liquefaction enzymes were often applied during mash preparation. In addition to the desired pectin hydrolysis activity, these enzymes also had pectin esterase activity, resulting in methanol formation of up to five to six times higher than in untreated fruit mash [17,51,52]. Such conventional, unspecific enzymes should only be used with caution—if at all—and only if methanol monitoring is implemented [51]. The use of commercial mash enzymes (i.e., pectolytic enzymes such as pectin methylesterase) always resulted in very high methanol contents (similar to the maximum methanol release potential) [25,53,54]. In the case of Rubinette apples, methanol increases between 5.5% and 12% occur after addition of various pectin enzymes, which are used to liquefy the mashes without adding water, compared to the untreated sample [34]. In quince, the lowest methanol contents were measured in the mashes blended with 33% water [25]. The avoidance of conventional liquefaction enzymes alone can lead to a 20% reduction in methanol content [47]. However, thick fruit mashes usually require a more or less high addition of water for fermentation and distillation, which means time and increased energy input during distillation, and at the same time leads to lower alcohol yields [34]. If pectinolytic enzymes have to be applied, pure lyases should be preferred (see Section 4.1.4). Besides the scrutiny in use of enzymes the raw material quality, mash preparation and fermentation conditions have potential to mitigate the methanol release.

#### 4.1.1. Quality and Treatment of Raw Materials

The methanol content is directly related to the fruit type or types used in the fermentation process (mainly dependent on the sugar/pectin ratio) but there are also differences between cultivars and harvest years [18,19,26,30]. For example, in studying distillates of Bartlett pear between 1978 and 1995, the 1993 vintage was the year with a strikingly lower methanol content [44]. In addition to the fruit type, it is very evident that the fruit quality used affects the quantity of the methanol formation [4,25]. At what stage of fruit development and how it is harvested also effects the methanol content [30].

Early harvest or hard pears led to higher methanol levels [34]. For pears and apricots, other researchers corroborated this finding showing that overripe fruit led to the lowest methanol contents [16]. In deviation of this finding, Adam reported an increase of methanol through advancing maturity of Williams Christ pears [44,47].

Utilization of plum juice leads to lower methanol contents than plum mashes [30]. On the other hand, destoned cherry mashes showed higher methanol contents than mashes with complete fruits including stones [55]. However, in another investigation of the same research group, destoned cherry mashes showed consistently lower methanol contents [56]. The conflicting results currently cannot be explained, other than confounding factors not controlled in the studies.

As pectins have a major occurrence in the skin layer, the removal of the fruit skins before fermentation may also reduce the methanol level by about 50% during production of wine spirits [31]. Cores and stems were also described to contain high levels of pectins [37]. Peeling and coring of pears, therefore, led to a methanol reduction of up to 42% [16]. However, this method is judged as not economically feasible for most spirits.

#### 4.1.2. Inhibition of Pectin Methylesterase by Acidification of Mash

pH is one of the most important factors which highly affects the activity of enzymes. Pectin methylesterase showed an optimum at pH 8 and 50 °C [57]. Other authors suggested pH 5–6 as optimum for pectin methylesterase [37,38]. Pectin methylesterases from yeast may have optimal pH values ranging from 3.75 to 6 [58].

Therefore, the proposed pH for fermentations to avoid pectin methylesterase activity is 2.5 [32,34] (Figure 1). No large differences were reported between pH 2.8 and 3.3, however [47]. Denes et al. [59] stated a decrease to 1% of the enzyme activity by decreasing the pH to 4.5 (pectin methylesterase from apples).

There is a clear indication from several studies of an up to 50% reduction in methanol by acidification of fruit mashes [4,22,25,26,33,34,35].

There is not a clear preference about the kind of acid to be used. Gössinger et al. suggest ortho-phosphoric acid (85%) [26,53] while Pieper et al. suggested sulfuric acid [35]. Commercially available products for acidification often contain mixtures of several acids such as malic acid/hydroxypropionic acid or phosphoric acid/lactic acid.

Gerogiannaki-Christopoulou used citric acid resulting in a decrease of about 15% methanol in grape pomace distillate [36]. However, while some organic acids such as citric acid might be depleted during fermentation by their inclusion in metabolic pathways, inorganic acids appear to be more appropriate. Buffer systems ensuring a long-term stability of mash pH might be an interesting option for future investigation.

#### 4.1.3. Inhibition of Pectin Methylesterase by Sterilization of Mash

A significant reduction of methanol by 40–90% [37,38] can be achieved by thermal deactivation of pectin methylesterase (often referred to as “mash heating”). There are various suggestions for temperature/time combinations to achieve the enzyme’s denaturation.

Sterilization at temperatures higher than 70 °C was generally suggested to effectively prevent the production of methanol by inactivation of pectin methylesterase [57,60]. Methanol can be reduced by targeted thermal deactivation of pectin methylesterase by heating the mash to 80 °C up to 85 °C for a holding time of 30 min or to 60 °C for 45 min [31,37,38]. Pasteurization at 72 °C for 15 s prevented the production of methanol in fermented plant beverages containing *Morinda citrifolia* (noni fruit) [60]. In cider spirit, the pasteurization (30 min at 50 °C, then heated to about 85 °C) of the apple juice prior to fermentation reduced the methanol content by 34–46% [18]. Lower methanol levels were obtained in Williams and plums by heating the mash to 65 °C for 5 min, followed by re-cooling for fermentation [34].

Xia et al. [22] confirmed that autoclaving by steam injection of the mash of jujube reduced the methanol content in the spirit significantly by a factor of about eight. The authors also determined pectin methylesterase activity confirming that their treatment method reduced the activity to one-fifth to half of that without treatment.

Further technological approaches for inactivation of methylesterase are thermosonication (ultrasound plus temperature at 70° led to 30% methanol reduction in plum wine) or use of microwaves (70 °C for 1 min led to 70% methanol reduction in plum wine). The authors indicated an additional nonthermal effect of both ultrasonication and microwaving with improved sensory properties of the product [41].

#### 4.1.4. Inhibition and Substitution of Pectin Methylesterase by Certain Additives

Pectinolytic enzymes (pectinase) are classified into esterase and depolymerase (lyase and hydrolase). Lyase produces oligo- or mono-galacturonate, while esterase produces pectic acid and methanol [61]. The addition of pectin lyase significantly (α = 0.01) reduced the resulting methanol contents in the mash of apricot and quince by 40–71% [25,26]. Lyase appears to inhibit the activity of the naturally contained pectolytic enzymes. The mechanism was speculated as being a cleavage of the pectin chains by the pectin lyase in such a fashion that the pectin fragments are not accessible as substrate for the pectin methylesterase [26]. The effectiveness of lyase enzymes can be increased by dilution of the mashes with water [26]. Similarly, the addition of certain detergents (anionic surfactants) as well as polyphenols (tannins) has a reducing effect on the release of methanol by full or partial inhibition of pectin methylesterases [19,34,35,41]. However, a large amount of agents is needed, which are rather expensive so that these methods were not widely implemented in practice [39].

Substituting the application of liquefying pectin methylesterase enzymes by pectinlysase reduced the methanol concentrations in apple distillates by 40–88%. The combination of mash sterilization (Section 4.1.3) and pectinlyase liquefaction resulted in an average methanol reduction of 94 ± 4% in the same distillates [48].

#### 4.1.5. Selection of Yeast Strains and Fermentation

Microbiological control of the process could also be used to prevent methanol formation in fermented beverages. For instance, pure culture inoculation using commercial yeast in contrast to spontaneous inoculation by wild yeasts should be practiced [43]. Mashes fermented without pure yeast cultures generally lead to higher methanol levels [34]. Yeast culture selection can reduce methanol contents in the distillates by up to 20% [34].

However, the reason why there are significant differences from yeast breed to yeast breed is hypothetically due to the fact that the individual breeds apparently differ in their ability to inhibit pectin esterase and thus the release of methanol from pectin [34]. Strains of *Saccharomyces* yeasts may produce all three types of pectinolytic enzymes (see Section 4.1.4) [61]. Selection of yeasts which do not form pectin methylesterase was suggested to contribute to reduction of methanol occurrence [33]. Selected mutant *Saccharomyces cerevisiae* S12 exhibited a methanol content during wine fermentations decreased by 73% compared to that of the wild-type strain [43]. On the other hand, Rodríguez Madrera et al. reported lower methanol concentrations in apple pomace spirits fermented with indigenous yeast than with commercial wine yeast [54].

In a comparison of three different yeast types (one newly developed strain with improved genetic and physiological performances and two commercial distillers’ yeasts), the new yeast showed higher methanol contents in plum and pear mashes, but not in cherry mashes [62]. In another investigation with the same yeast types, the new yeast showed lower methanol contents in plum mashes but higher in cherry mashes [55]. In a third study with these yeast types, the new yeast showed consistently lower methanol values than the commercial yeast in cherry spirits [56]. These conflicting results were interpreted by other influences on methanol content rather than a yeast influence. Similarly, different strains of yeast were used in fermentations but no significant change in the quality or quantity was noticed over time [4].

Another microbiological method for the control of methanol in fermented beverages, might be the use of methylotrophic yeast such as *Pichia methanolica* [63] and *Candida boidinii* [64] which have the capacity of utilizing pectin or the methyl ester moiety of pectin and methanol, thus preventing the accumulation of methanol in fermented products [61]. However, the application of these microorganisms for fermentation of spirits has not been demonstrated so far.

#### 4.1.6. Fermentation Conditions

The activity of the pectin methylesterase enzyme is directly linked with the temperature [65]. Increasing the temperature of the mash increases the speed of reaction until the temperature reaches a very high level where the enzyme starts denaturizing (see Section 4.1.3). Lowering the fermentation temperature from 20 °C to 12 °C with use of cold fermentation yeast may result in a 10–24% reduction in methanol release in the mash [26], but not in all cases [25,26].

### 4.2. Storage of Fermented Mash before Distillation

Generally, the storage time following fermentation has a major influence on the methanol release (Figure 1) [32]. Depending on the pH level, an almost 100% release can be expected after only some weeks of storage. During mash storage of 4 weeks, methanol contents increased, in some cases sharply by 15–50% [25,26]. Therefore, the optimal practice would be to conduct the distillation as soon as fermentation has been complete or at least to minimize storage time as far as possible [33].

### 4.3. Distillation Method and Conditions

#### 4.3.1. Methanol Reduction during Pot Still Distillation

Methanol has a boiling point (64.7 °C) that is considerably lower than the ones of ethanol (78.5 °C) and water (100 °C). However, it is nevertheless difficult to separate methanol from the azeotropic ethanol-water mixture [14]. When the alcohol mixture is distilled in simple pot stills such as the ones used by most small-scale artisanal distilleries throughout Central Europe, the solubility of methanol in water is the major factor rather than its boiling point. As methanol is highly soluble in water, it will distil over more at the end of distillations when vapours are richer in water. That means, methanol will appear in almost equal concentration in almost all fractions of pot still distillation in reference to ethanol (i.e., as g/hL pa), until the very end where it accumulates in the so-called tailings fraction (Figure 2) [4,5,14,20,32,37,40,47]. However, even today many professional distillers believe that methanol concentrates preferably in the first fractions (heads fractions). And that methanol is the reason that heads fractions smell and taste bad (which is caused by acetaldehyde and ethyl acetate but not by methanol). It is of note that single studies that suggested that methanol may be enriched in the first distillation fractions were not plausible and potentially erroneous (e.g., compare the abstract with the conclusion section in Xia et al. [22], which report completely conflicting information—from the data presented in the work it can be assumed that the study from China is in fact corroborating the studies from Europe and the United States that methanol is enriched in the tailings while the information in the abstract that it is enriched in the heads fractions is most probably a translation mistake).

Various distillation tests carried out show that the methanol content in the product (hearts) fractions can hardly be influenced by different distillation techniques. Even in experiments with various “catalysts”, no groundbreaking findings have yet emerged. Only relatively expensive silver wool as adsorbent led to methanol reductions of up to 20% [34].

Therefore, the separation of tailings, which also has to be done for sensorial reasons, is so far the only option for a reduction of methanol during pot still distillation. The reduction of methanol contents of the product fractions in g/hL pa compared to mash may be between 20 and 30%. On the other hand, an extremely late separation of tailings can cause an increase of methanol contents of about 20% in the product fractions [39].

In general, it can be seen that the methanol content in the spirit increases with reflux ratio increases. That means the higher the reinforcement and the slower the distillation is, the higher the methanol content in the distillate [32,66] (Figure 2). Distillation parameters also had an influence on the methanol content of the distillates. Especially the dephlegmator temperature showed a significant effect on the methanol content. Within the parameters tested using 150 L still, three trays and one dephlegmator, the decrease in methanol content varied between 16% and 36% [25].

On the other hand, Scherübel [20] suggests the following three measures to reduce methanol by improvements in pot still distillation:Perform double distillation: it is always advisable to carry out two subsequent distillations with regard to methanol separationIncrease separation efficiency: The methanol separation can be increased by a simple optional parallel connection of a conventional spirits tube and a more separation-efficient column. If possible, this column should be at least partially cooled at the top to increase internal reflux and thus separation efficiency.Cooling at the head: When use of an additional column is not feasible, partial cooling of the spirits tube at the beginning of the second distillation can also increase the internal reflux and thus increase the separation efficiency.

In summary, there is still a bit of discrepancy regarding the influence of reflux ratios between the different studies in the literature. This can probably be explained by the wide variability of commercially available stills and legal differences (number of plates) for artisanal distilleries in different jurisdictions.

#### 4.3.2. Methanol Reduction during Large-Scale Distillation

In contrast to pot stills that typically consist of a small column (three or four plates), industrial-scale distilleries with 15 to 30 plates provide the possibility of continuous distillation and advanced regulation of distillation including processes of demethylation [39].

Methanol content can be decreased during the rectification by using demethanolization columns [33,40]. This process is efficient and successfully reduces the methanol content up to 40–90% in comparison to the starting amount. However, investment is only viable for rather big businesses with high capacity utilization [39].

A combined evaporation/condensation method to reduce methanol from distillates was patented by Capovilla [46]. The application of the method was found to reduce methanol in fruit spirits by 58–190 g/hL pa [42]. However, such methods may not be economically viable as they considerably reduce the alcohol content along with the methanol content [26]. The promised results of the evaporation/condensation method were also criticized as implausible with independent investigations showing lesser methanol reduction (9–92 g/hL pa) always connected with inacceptable losses of ethanol (up to 10% vol) [45]. All in all evaporation/condensation methods for demethanolization were judged as economically unviable specifically for smaller businesses.

### 4.4. Storage of Distillate after Fermentation

Not much evidence is available regarding the methanol evolution during the distillates’ storage and aging process. Botelho et al. [4] suggested a tendency for low amounts of methanol in advanced wood-cask aged spirits, attributable to methanol oxidation and subsequent acetalization reaction with the formation of diethoxymethane. On the other hand, methanol is expected to be quite stable in inert containers without the presence of oxygen. This is also in line with the authors’ experience from validating methods for methanol determination, which suggested that methanol is a stable compound in bottled hydroalcoholic solutions [67].

Similar results were observed by Xia et al. [22]. The 270-day storage of jujube spirit in oak barrels significantly reduced its methanol content, while lower reductions were observed in plastic or stainless-steel containers. The authors explained the reduction by esterification reactions but were unable to provide explanation for the differences between container materials.

## 5. Discussion

### 5.1. Good Manufacturing Practice for Methanol Reduction Leading to Decreased Levels in Commercial Products

The only currently available review about methanol reduction possibilities has been provided by Botelho et al. [4] in the context of a more general review on quality of fruit spirits. While being less comprehensive and lacking the coverage of major studies only available in German language, the major areas influencing the methanol content in fruit spirits were in agreement with this review, namely, raw material quality, fermentation, storage, and distillation. Botelho et al. [4] concluded that the reduction of the time between fermentation and distillation being the most effective way to reduce the methanol content of the final beverage, with that suggestion to be classified as “good manufacturing practice”. This is also in agreement with the comprehensive book of Adam and Versini published by the European Commission [39].

The quality of the raw material used is a key factor which defines the quality of the spirit produced and its methanol content. Alcoholic beverages derived from materials low in pectin content (such as beer, wine or grain-based spirits such as whiskey) have typically a much lower concentration of methanol than fruit-based products. Mitigation efforts in the past were therefore focused on fruit spirits.

Previous results have shown that industry efforts and application of improved fermentation and distillation technology have led to lowered methanol levels in fruit spirits [1]. Due to the limits for methanol introduced uniformly throughout Europe in 1989, processes were developed to reduce this substance in spirits [1]. Methanol release during fermentation and distillation is not a univariate process, but a combination of several measures can effectively ensure methanol levels below legal limits.

According to Glatthar et al. [32], the following mitigation measures are simple and can be applied even by small, artisanal distilleries:Adjust the mash pH before fermentation to pH 2.5–3.0Short fermentation using inoculation with yeasts followed by immediate distillationDo not recycle the tailings

Using these measures, a methanol content reduced by half, without changing the sensory quality of the products, can be expected.

Interestingly, all the measures discussed before may have led to considerably decreased levels of methanol in commercial products on the European market and can be seen as an excellent example of implementation of research results into practice. This may be evidenced by the efforts of the researchers to publish their results in addition to the usual peer-reviewed journals in trade journals in a format readable and understandable by distillers.

Adam and Postel [68] showed that cherry brandies tested in 1991 had almost 100 g/hL pa less methanol than cherry brandies produced before 1986. Adam and Versini [39] confirmed this trend in 1996. Own investigations of 923 cherry spirits (Figure 3), which is one of the most frequently tested product groups at the CVUA Karlsruhe as this product is traditionally a specialty of North Baden or the Black Forest, analyzed during the years 1980–2020 confirm a statistically significant linear decrease in methanol content (r = −0.345, *p* < 0.0001). Mean methanol contents decreased from an average of 500 g/hL pa in the early 1980s to an average of 400 g/hL pa at present (for methodology and details on samples 1980–2003, see [1]). None of the samples was found to exceed the EU limit of 1000 g/hL pa.

### 5.2. Coffee Spirits—A Special Case for Methanol Mitigation

Despite some anecdotal evidence that spirits derived from coffee cherries or coffee by-products were traditionally manufactured in some coffee-producing countries such as Nepal, there is not only extremely limited evidence on production methods [69] but also on chemical composition and specifically the methanol content of coffee cherry spirits. Especially the coffee pulp juice from wet-processing with about 3–5% of total sugars is an adequate substrate for production of ethanol [70]. For coffee mucilage from various *Coffea arabica* varieties, the pectin yield in the coffee fruit was 0.03–0.09% and methoxyl esterification degrees of 19–31% were reported [71]. Coffee pulp of *Coffea canephora* contains 2–3% pectin with a methoxyl esterification degree of about 6% [72], while higher contents were reported for *Coffea arabica* with 15% pectin in dried pulp with a methoxyl esterification degree of 63% [73]. Another study reported 11% pectin in *Coffea arabica* without specifying the esterification degree [74].

Depending on species and processing, the pectin content of *Coffea* by-products could be higher than the one in most other fruits used for spirits production, such as cherries (0.4%), apricots (1%), or apples (0.8%) [75], while the methoxyl degree of *Prunus avium* cherries was between 44% and 91% depending on extraction method and ripening stage [76]. Hence, the capacity for enzymatic methanol formation may be higher in coffee cherries than in conventional materials for fruit distillate production. From the few studies on spirits produced from coffee cherries or coffee-by products many did not investigate methanol contents [77,78,79,80,81], which is a bit puzzling because methanol is typically included in any standard spirits analysis [82].

Nevertheless, there are some studies on methanol in fermentations of coffee materials available (Table 2). Bonilla-Hermosa et al. [74] showed comparably low levels of methanol in coffee pulp mixed with coffee wastewater from the depulping and demucilage process of *Coffea arabica* beans. However, only the fermentation mash was analyzed in this case and no distillation was conducted. A study of spent coffee grounds fermented with added sugar of 180 g/L also showed rather low levels of methanol. On the other hand, Somashekar and Appaiah [83] showed that solid substrate fermentation of coffee cherry husk from *Coffea canephora* with *Clavispora* and *Pichia* strains may lead to considerable levels of 7.2–10.8% methanol. The process was intended for technical alcohol production and appears as completely unsuitable for obtaining products for human consumption. While the production of technical alcohols from coffee by-products and waste-products could be an interesting valorization option, this study shows that extreme scrutiny has to be applied if spirits from coffee by-products are intended to be used as consumer products.

This concern was strengthened by the informative pilot study of Einfalt et al. [15] reporting results of coffee cherry spirit production. The mash was prepared using *Coffea arabica* cherries transported in frozen form from Thailand to Germany, where they were pulped. After lowering the pH to 3.1 using phosphoric acid/lactic acid addition, a commercial pectinase enzyme was added for liquefaction. After addition of commercial yeast, the mash was distilled after 17 days of fermentation. The methanol content in the hearts fractions was 2600 ± 400 g/hL pa, which considerably exceeded the EU limit of 1000 g/hL pa and offers a safety margin of less than 2 for the level of acute toxicity of 5000 g/hL pa (see Section 3). The authors suggested that the application of pectinase and the long storage had an adverse effect on the methanol concentration [15], which is plausible considering our review results in Section 4.1 and Section 4.2.

In a patented method by Bodmer and Ruder [84], whole coffee cherries were mashed with addition of 5% sugar, adjusted to pH 3.0 with phosphoric acid/lactic acid, and pitched with *Saccharomyces cerevisiae* yeast and diammonium phosphate. After a fermentation time of 7–14 days, the mash was double-distilled using pot still technique. The methanol contents of two coffee cherry spirits were 684 and 573 g/hL pa. While the production method with addition of sugar is not compliant with the EU regulation for fruit spirits, where the products’ ethanol must exclusively originate from fruits [23], this also lowers the relative methanol content by increasing the ethanol content. Hence, it can be deduced from the results that a coffee cherry spirit production according to the patented method, excluding artificial sugar addition, would lead to a methanol limit exceedance similar to the results of Einfalt et al. [15].

In conclusion, apart from the lack of novel food approval [85], none of the coffee cherry spirits presented so far would have been compliant with the EU spirits regulation. It is clearly necessary to apply the gathered knowledge about methanol mitigation possibilities in further research of this interesting novel type of spirit, so that compliant coffee cherry spirits will hopefully be available in the future.

**Table 2 molecules-26-02585-t002:** Methanol content in spirits produced from coffee cherries and coffee by-products.

Raw Material	Methanol Content	Compliance with EU Regulation for Fruit Spirits ^1^	References
Coffee cherry	2600 ± 400 g/hL pa	no ^2^	[15]
Coffee cherry + 5% sugar	573–684 g/hL pa	no ^3^	[84]
Coffee cherry husk	7–11%	(non-food product)	[83]
Coffee pulp mixed with coffee wastewater (1:10)	40–128 μg/L (in mash)	(no distillation conducted) ^4^	[74]
Spent coffee grounds + 18% sugar	11 ± 3 mg/L (44 ± 12 g/hL pa ^5^)	no ^4^	[86]

^1^ This does not suggest general compliance with EU food regulations. Novel food approval is needed in the EU for most coffee by-products and derivative products before being placed on the market [85]. ^2^ Exceedance of general methanol limit for fruit spirits of 1000 g/hL pa [23]. ^3^ The production method with added sugar is not compliant with EU regulations for fruit spirits; without sugar addition, the methanol limit of 1000 g/hL pa would have likely been exceeded. ^4^ Fruit sprit ethanol must exclusively originate from fresh fruits [23] and not from waste products such as spent coffee grounds or wastewater. Potential compliance in another spirit drinks’ category or as a generic ‘spirit drink’ needs to be checked. ^5^ Recalculation (alcoholic strength at 40% vol).

## 6. Conclusions

The methanol content is among the key parameters for determining the regulatory compliance of spirits and other alcoholic beverages. The mitigation measures developed over the last decades allowed industry not only to conform to the EU standards but also to increase the margin of safety by generally lowering the methanol content in the category of fruit spirits.

Interestingly, coffee cherry pulp, which is produced in large quantities as a by-product of coffee manufacturing, was proposed as a material to produce spirits. Very high concentrations of methanol were found in coffee cherry spirit compared to other fruit spirits. Hence it is specifically necessary to mitigate the methanol content in these spirits to uphold the legal requirements and to protect public health from this potential hazard.

## Figures and Tables

**Figure 1 molecules-26-02585-f001:**
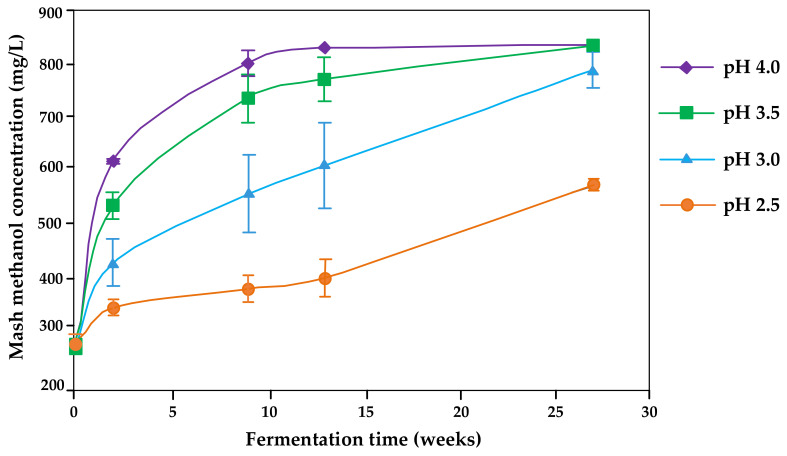
Kinetics of methanol formation in Bartlett pear mashes affected by the initial mash pH and fermentation time (redrawn from [32]).

**Figure 2 molecules-26-02585-f002:**
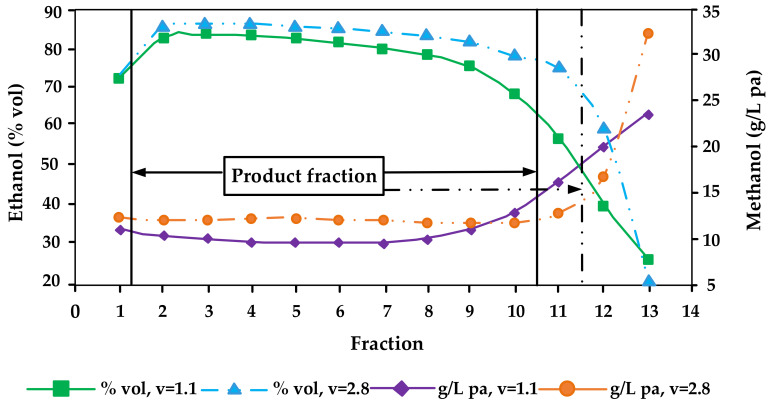
Distillation characteristics of ethanol and methanol affected by different reflux ratios (v) during distillation of Bartlett pear mashes (redrawn from [32]).

**Figure 3 molecules-26-02585-f003:**
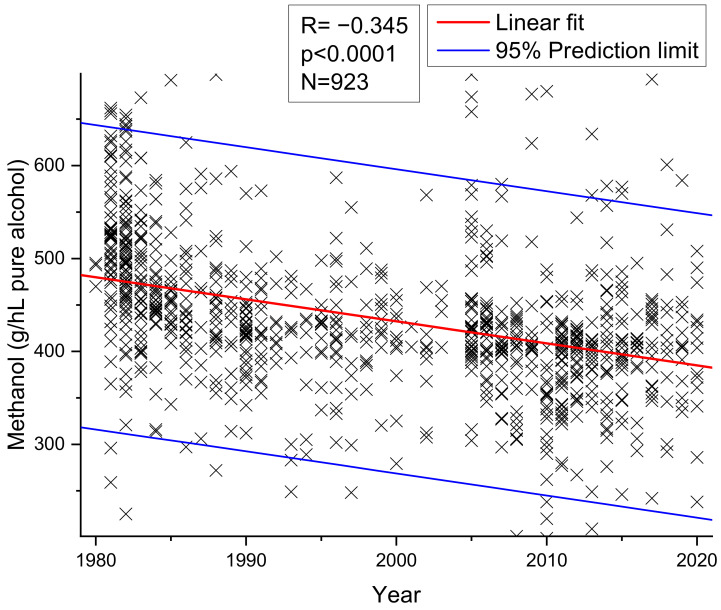
Methanol contents of 923 cherry spirits analysed between 1980 and 2020.

**Table 1 molecules-26-02585-t001:** Summary of major methods to reduce methanol during production of fruit spirits.

Method	Methanol Reduction Potential ^1^	Authors’ Judgment about Applicability	References
Improvement of quality of raw material	up to 40%	Raw material is extremely important and the type and quality highly affects the methanol content. Removal of pectin-rich fruit parts such as skins may reduce methanol content.	[4,16,25,30,31]
Acidification of mash	up to 50%	Acidification of mash inhibits the activity of pectin methylesterase. It also inhibits spoilage microorganisms, which may produce pectin methylesterase.	[25,32,33,34,35,36]
Sterilization of mash	40–90%	Temperature treatment efficiently denaturizes pectin methylesterase enzymes. High energy requirement and not feasible for artisanal distillers.	[18,22,31,37,38,39,40,41]
Decreased storage time of fermented mash before distillation	up to 50%	Storage time should be avoided or being minimized as far as possible, because sharp methanol increases were reported during storage.	[26,32,33]
Selection of appropriate yeast strains	up to 25%	Yeasts with low capacity of producing pectin methylesterase to be preferred.	[4,30,42,43]
Decreased fermentation temperature	up to 25%	Lower temperatures and the use of cold fermentation yeast is recommended.	[26]
Improvement in distillation method and conditions	up to 80%	Methanol is enriched in tailings. Earlier cut (not below 50% vol). No recycling of tailings.	[4,14,20,22,30,32,33,34,44]
Demethanolization following distillation	50–90%	Effective in industry but not feasible for small artisanal distillers, high expenditure	[39,40,42,44,45,46]
Avoidance of liquefaction enzymes	up to 20%	Avoid pectin methylesterase enzymes which release methanol.	[4,22,26,34,39,42,47]
Application of alternative liquefaction enzymes	up to 88%	Substitute pectin methylesterase enzymes by pectin lyase enzymes to reduce the release of methanol	[48,49]

^1^ Authors’ estimation if several studies were available.

## Data Availability

No new data were created or analyzed in this study. Data sharing is not applicable to this article.

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
