# Peer review of "Methanol Mitigation during Manufacturing of Fruit Spirits with Special Consideration of Novel Coffee Cherry Spirits"

_molecules, 2021, doi:10.3390/molecules26092585_

Round 1

Reviewer 1 Report

This is a good work that evaluates the possibilities to control and reduce the methanol content in fruit spirits and describes some initial observations for the novel spirit made from coffee cherries. I have examined the submitted paper very carefully and I recommend its publication after change the next,  Instead of “XXX°C“, should be “ XXX °C”. 

Author Response

This is a good work that evaluates the possibilities to control and reduce the methanol content in fruit spirits and describes some initial observations for the novel spirit made from coffee cherries. I have examined the submitted paper very carefully and I recommend its publication after change the next,  Instead of “XXX°C“, should be “ XXX °C”. 

Response: Thank you for your evaluation! We have added a space before °C throughout.

Reviewer 2 Report

Dear Editor and Authors,

The manuscript ‘Methanol mitigation during manufacturing of fruit spirits with  special consideration of novel coffee cherry spirits’ by Patrik Blumenthal, Marc C. Steger, Daniel Einfalt , Jörg Rieke-Zapp , Andrès Quintanilla Bellucci , Katharina  Sommerfeld , Steffen Schwarz , and Dirk W. Lachenmeier is a very interesting review on possibilities of methanol reduction in spirits. Very well written. The Authors planned to use coffee cherries as a substrate for production of spirits. Unfortunately, the coffee cherries are high in pectins and produce a high quantity of methanol during distillation of ethanol in the spirits. The Authors reviewed the possibilities to reduce methanol in fruit spirits

including the pH value of the mash, the addition of various yeast and enzyme preparations, fermentation temperature, mash storage, and most importantly the raw material quality and hygiene. The Authors found lowering the pH value and the use of cultured yeasts when mashing fruit substances as best practice common in industry. The review is useful for spirit industry but also of high scientific value. I recommend acceptance of the manuscript,

Yours sincerely,

Author Response

Dear Editor and Authors,

The manuscript ‘Methanol mitigation during manufacturing of fruit spirits with  special consideration of novel coffee cherry spirits’ by Patrik Blumenthal, Marc C. Steger, Daniel Einfalt , Jörg Rieke-Zapp , Andrès Quintanilla Bellucci , Katharina  Sommerfeld , Steffen Schwarz , and Dirk W. Lachenmeier is a very interesting review on possibilities of methanol reduction in spirits. Very well written. The Authors planned to use coffee cherries as a substrate for production of spirits. Unfortunately, the coffee cherries are high in pectins and produce a high quantity of methanol during distillation of ethanol in the spirits. The Authors reviewed the possibilities to reduce methanol in fruit spirits including the pH value of the mash, the addition of various yeast and enzyme preparations, fermentation temperature, mash storage, and most importantly the raw material quality and hygiene. The Authors found lowering the pH value and the use of cultured yeasts when mashing fruit substances as best practice common in industry. The review is useful for spirit industry but also of high scientific value. I recommend acceptance of the manuscript,

Yours sincerely,

Response: Thank you for the assessment of our manscript! We are glad that our manuscript is seen as helpful for industry and hope that the trend of reduced methanol contents may continue, providing an increased safety margin for the consumer.

Reviewer 3 Report

Blumenthal and collegues presented a review regarding the main approaches (raw material, pH, microbiological etc.) used to reduce the methanol content during the production of fruit spririts. In addition, the authors also covered the issue regarding the mitigation of methanol in spirits obtained from coffee cherry.  

The latter products have been treated separately because they are distinguished from the others (fruit spirits) by a higher methanol content.

The topics covered are interesting and well described, however I would just have some considerations regarding the section of discussion.
In my opinion, the discussion is too concise and does not deal in depth with all the issues mentioned in the previous paragraphs (eg. cherry spirit with coffee). It would be advisable to integrate and better argue this section of the review.

Author Response

Blumenthal and colleagues presented a review regarding the main approaches (raw material, pH, microbiological etc.) used to reduce the methanol content during the production of fruit spirits. In addition, the authors also covered the issue regarding the mitigation of methanol in spirits obtained from coffee cherry.  

The latter products have been treated separately because they are distinguished from the others (fruit spirits) by a higher methanol content.

The topics covered are interesting and well described, however I would just have some considerations regarding the section of discussion.
In my opinion, the discussion is too concise and does not deal in depth with all the issues mentioned in the previous paragraphs (eg. cherry spirit with coffee). It would be advisable to integrate and better argue this section of the review.

Response: Thank you for suggesting the lack of flow within the discussion and the remark about coffee spirits, which might be better fitting in the discussion, as these observations are basically outside the scope of the review on methanol reduction. We have changed the flow of the sections and integrated the part on coffee spirits at the end of the discussion. Please note that this led to considerably re-numbering of references. We also detected a reference that was cited twice (Liang et al.), which was now corrected. This explains the change in total number of references.

Reviewer 4 Report

This paper addresses the problem of methanol occurence in alcoholic bevarages with the special focus on coffee cherry spirit. The paper is written in a logical and systematic way and can be accepted in the present form.  

Author Response

This paper addresses the problem of methanol occurrence in alcoholic beverages with the special focus on coffee cherry spirit. The paper is written in a logical and systematic way and can be accepted in the present form.  

Response: Thank you for reviewing our paper. Please note that due to the suggestion of reviewer #3 we have changed the flow of the paper, and hope that we now have even increased the systematic logic of our argumentation.